# The Effect of Acidic and Alkaline Seawater on the F-Actin-Dependent Ca^2+^ Signals Following Insemination of Immature Starfish Oocytes and Mature Eggs

**DOI:** 10.3390/cells12050740

**Published:** 2023-02-25

**Authors:** Nunzia Limatola, Jong Tai Chun, Suzanne C. Schneider, Jean-Louis Schmitt, Jean-Marie Lehn, Luigia Santella

**Affiliations:** 1Department of Research Infrastructures for Marine Biological Resources, Stazione Zoologica Anton Dohrn, 80121 Napoli, Italy; 2Department of Biology and Evolution of Marine Organisms, Stazione Zoologica Anton Dohrn, 80121 Napoli, Italy; 3Laboratory of Supramolecular Chemistry, Institut de Science et d’Ingénierie Supramoléculaires ISIS, Université de Strasbourg, 8 Allée Gaspard Monge, 67000 Strasbourg, France

**Keywords:** maturation, starfish, pH, acrosome reaction, Ca^2+^ signaling, jelly coat, fertilization, polyspermy

## Abstract

In starfish, the addition of the hormone 1-methyladenine (1-MA) to immature oocytes (germinal vesicle, GV-stage) arrested at the prophase of the first meiotic division induces meiosis resumption (maturation), which makes the mature eggs able to respond to the sperm with a normal fertilization response. The optimal fertilizability achieved during the maturation process results from the exquisite structural reorganization of the actin cytoskeleton in the cortex and cytoplasm induced by the maturing hormone. In this report, we have investigated the influence of acidic and alkaline seawater on the structure of the cortical F-actin network of immature oocytes of the starfish (*Astropecten aranciacus*) and its dynamic changes upon insemination. The results have shown that the altered seawater pH strongly affected the sperm-induced Ca^2+^ response and the polyspermy rate. When immature starfish oocytes were stimulated with 1-MA in acidic or alkaline seawater, the maturation process displayed a strong dependency on pH in terms of the dynamic structural changes of the cortical F-actin. The resulting alteration of the actin cytoskeleton, in turn, affected the pattern of Ca^2+^ signals at fertilization and sperm penetration.

## 1. Introduction

The landmark discovery of 1-methyladenine (1-MA) as the hormone inducing the resumption of the meiotic cycle (maturation) of immature starfish (GV-stage) oocytes arrested at the first prophase of the meiotic division has allowed profound in vitro studies on the structural and biochemical changes that immature oocytes must undergo to reach the state of optimum fertilizability following the maturation process [1,2]. It has been reported that 1-MA activates a yet unknown receptor that releases the βγ subunits of its coupled heterotrimeric G-protein from the α subunit upon hormonal stimulation. The interaction of βγ subunits with effectors leads to an intracellular pH increase and activation of cyclin B-Cdk1, which is known as MPF (Maturation Promoting Factor) inducing GVBD (Germinal Vesicle Breakdown). The latter events are required for the spindle assembly at metaphase I [3,4]. The 1-MA-induced morpho-functional changes in the oocyte are necessary to make the mature eggs competent to respond to the activating sperm with a normal fertilization response and subsequent embryonic development. The possibility of inducing meiotic maturation in vitro provides an advantage for starfish oocytes as an experimental model. Furthermore, at variance with sea urchin eggs that are spawned and fertilized in seawater after the second maturation division, starfish oocytes can be inseminated before, during, and after the course of the maturation process to study the nature of the altered physiological conditions leading to polyspermic fertilization [5,6,7,8].

Previous studies have highlighted the importance of the structural dynamics of the cortical F-actin network following hormonal stimulation of immature oocytes [9] in eliciting Ca^2+^ signals in the cytoplasm and nucleus and at different stages of the maturation process [10,11]. Specifically, the two modes of Ca^2+^ increases taking place a few minutes after hormonal stimulation and during GVBD have underscored the contributions made by the state of the cortical actin cytoskeleton and by intermixing the nuclear and cytoplasmic components in producing mature eggs apt for normal fertilization [11,12]. Indeed, immature oocytes whose surface topography and structural organization of the oocyte cortex are strikingly different from mature eggs experience multiple Ca^2+^ waves at fertilization due to polyspermy [7,8]. In fact, numerous sperm are incorporated into the GV-stage oocytes, but they still remain localized in the cortical region without undergoing structural changes such as DNA decondensation and the formation of aster and pronucleus due to their cytoplasmic “immaturity” [13].

It has been shown that the 1-MA-dependent maturation of the intracellular Ca^2+^ stores enables a higher Ca^2+^ response to inositol 1,4,5-trisphosphate (InsP_3_) in mature starfish eggs and the fertilizing sperm. The sensitization of the Ca^2+^ stores to InsP_3_ might be attributed to the structural changes in the endoplasmic reticulum during the maturation process of the oocytes [14,15]; however, it was then suggested that it was linked to the remodeling of the MPF-dependent actin cytoskeleton in the cortex and cytoplasm induced by the 1-MA [16,17,18]. Notably, the F-actin-dependent morphological modifications of the cortex of maturing oocytes also include the shortening of microvilli (filled with actin filaments) on the plasma membrane of mature eggs [19,20,21]. This microvillar morphology restructuring leads to changes in plasma membrane resting potential, which constitute an essential physiological condition for a normal electrophysiological and Ca^2+^ response upon insemination of mature eggs [11,21,22,23], in line with the idea of a cytoskeletal modulation of Ca^2+^ signals in the cells mediated by microvilli [24,25].

Following insemination, a mature starfish egg experiences electrical and Ca^2+^ responses due to its interaction with the tip of the acrosomal process (approximately 20 µm long) formed on the sperm head when it comes into contact with the jelly coat (JC) surrounding the egg [26,27,28,29]. The first detectable Ca^2+^ signal occurs in the form of a fast (1-to 3 sec) subitaneous Ca^2+^ increase in the periphery of the egg cortex (the cortical flash, CF), which is followed by a Ca^2+^ wave starting at the site of sperm-egg interaction and propagating to the opposite pole [20,21,23,30,31,32,33].

The findings from the starfish mature eggs on the morpho-functional importance of the cortical actin filaments in inducing a normal Ca^2+^ response and monospermic penetration have been extended to *Paracentrotus lividus* sea urchin eggs [34]. Recent studies have highlighted the essential roles played by the integrity of the vitelline layer tightly bound to microvilli and the F-actin-associated cortical granules and vesicles by showing that microvillar morphology changes and disruption, relocation, or fusion of the cortical vesicles all lead to abnormal fertilization responses [35,36,37]. Confirmation of a crucial role of the structural morphology of microvilli and egg cortex in achieving a normal Ca^2+^ signal and monospermic fertilization in sea urchin eggs has been provided by showing that the alteration of the egg cortex induced by exposure to unfertilized *P. lividus* eggs to seawater containing low sodium or an acidic pH (pH 6.8) compromised the sperm-induced Ca^2+^ signals and embryo development [38,39]. A more recent study on the effect of the incubation and fertilization of *P. lividus* eggs in acidic (pH 5.5) and alkaline seawater (pH 9) has highlighted the strict interdependence between seawater pH and the fertilization response due to the changes in the morpho-functional alterations of the F-actin of the cortex of unfertilized eggs. Specifically, the acidic and alkaline pH of the seawater evoked depolymerizing and polymerizing effects on cortical F-actin, respectively, and this, in turn, influenced the Ca^2+^ response at fertilization and sperm entry [37]. Finally, the elongation of microvilli [40] and F-actin spikes formation in the perivitelline space [41] as well as translocation of actin fibers from the zygote surface [42], sperm incorporation, and cleavage [38] have sanctioned the indispensable role of the egg cortical F-actin structure and its remodeling in the regulation of all the aspects of the fertilization process.

The present contribution has analyzed the effect of the acidic and alkaline seawater pH on the response of immature oocytes of starfish to insemination. It is well known that the GV-stage (immature) oocytes respond to sperm addition by eliciting multiple Ca^2+^ signals due to polyspermic interactions [20,21,43]. Polyspermy probably arises from the different organization of the oocyte’s surface and the actin cytoskeleton of their cortex compared to mature eggs [8]. Thus, our experimental design focused on how the acidic or alkaline conditions would affect the structural F-actin dynamics and polyspermy in immature oocytes as well as the process of meiotic maturation and the pattern of the fertilization response.

## 2. Materials and Methods

### 2.1. Gametes Collection, Maturation, and Fertilization In Vitro

*Astropecten aranciacus* (starfish) were collected from the end of January to May in the Gulf of Naples and Gaeta and maintained at 16 °C in circulating seawater. GV-stage oocytes were isolated from the ovary by making a small hole in the dorsal region of the female animal. The oocytes were sieved with gauze and collected in natural seawater (NSW, pH 8.1) filtered with a Millipore membrane of 0.2 µm pore size (Nalgene vacuum filtration system, Rochester, NY, USA). The collected oocytes kept in NSW were used for maturation and fertilization experiments within the next 3 hr. The dry sperm collected from the testis were diluted in NSW and used to inseminate oocytes and eggs at a final concentration of 1 × 10^6^ sperm/mL. In vitro maturation was performed by adding the hormone 1-methyladenine (1-MA) (Acros Organics, Fisher Scientific, Milan, Italy) to the GV-stage oocytes suspended in NSW at different pH at a final concentration of 10 μL/mL. The lowering of pH was obtained immediately before the experiment by adding HCl to NSW (pH 8.1) until reaching the needed value (pH 6.8) [39]. The pH of seawater was raised to 9.0 by adding sufficient NH_4_OH to NSW (pH 8.1), about 1 mmol of NH_4_OH/l, which was previously used in studying membrane potential and cortical F-actin structural changes upon the exposure of sea urchin eggs to NH_4_OH seawater [40,44].

### 2.2. Light Microscopy and Transmission Electron Microscopy (TEM)

Light microscopy was performed with the Leica DMI6000 B system to monitor the surface and cortical changes following the insemination of immature oocytes and mature eggs in acidic or alkaline NSW at different time points. For TEM analyses, immature oocytes and oocytes matured and fertilized in acidic or alkaline seawater were fixed in NSW containing 0.5% glutaraldehyde (pH 8.1) for 1 h at room temperature. After extensive washing in NSW, the samples were post-fixed with 1% osmium tetroxide and 0.8% K_3_Fe(CN)_6_ for 1 h at 4 °C. The samples were washed in NSW and rinsed with distilled water (3 times, 10 min each), and finally treated with 0.15% tannic acid for 1 min at room temperature. The specimens were then dehydrated in ethanol with increasing concentrations. Residual ethanol was removed with propylene oxide before embedding in EPON 812. Ultrathin sections were made with the ultramicrotome Leica EM UC7 (Leica Microsystems, Wetzlar, Germany) and observed under a Zeiss LEO 912 AB (Carl Zeiss Microscopy Deutschland GmbH, Oberkochen, Germany) without staining.

### 2.3. Microinjection, Ca^2+^ Imaging, Fluorescent Labeling of F-Actin and Extracellular Matrix

Intact immature and maturing oocytes were microinjected using an air-pressure transjector (Eppendorf Femto-Jet, Hamburg, Germany). To monitor intracellular Ca^2+^ level changes in the GV-stageoocytes and mature eggs (treated with 1-MA for 70 min) were microinjected 10 min before hormonal stimulation or insemination with 500 µM Calcium Green 488 conjugated with 10 kDa dextran mixed with 35 µM Rhodamine Red (Molecular Probes, Eugene, OR, USA) in the injection buffer (10 mM Hepes, 0.1 M potassium aspartate, pH 7.0). The fluorescence images of cytosolic Ca^2+^ were captured with a cooled CCD camera (Micro-Max, Princeton Instruments Inc., Trenton, NJ, USA) mounted on a Zeiss Axiovert 200 with a Plan-Neofluar 40/0.75 objective at about 2 s intervals, and the data were analyzed with MetaMorph (Universal Imaging Corporation, Molecular Devices, LLC, San Jose, CA, USA). Following the formula, F_rel_ = [F−F_0_]/F_0_, where F represents the average fluorescence level of the entire egg and F_0_ the baseline fluorescence, the overall Ca^2+^ signals were quantified for each moment, and F_rel_ was expressed as RFU (relative fluorescence unit) for plotting the Ca^2+^ trajectories. Applying the formula F_inst_ = [F_t_−F_(t−1)_]/F_(t−1)_, the instantaneous increment of the Ca^2+^ level was analyzed to locate the specific area of transient Ca^2+^ increase. The values of Ca^2+^ signals were obtained from three independent experiments (N) using three different females, and the number of eggs (n) being analyzed for each condition is specified in the Results. To visualize the actin cytoskeleton, 50 µM (pipette concentration in methanol) of the fluorescent F-actin probe AlexaFluor 568-phalloidin (Molecular Probes, Thermo Fisher Scientific, Oregon, USA) was microinjected into GV-stage oocytes and the unfertilized mature eggs in two independent experiments (N = 2), using two different females. To study the effect of the altered pH of the seawater on the morpho-functionality of the jelly coat triggering the sperm acrosome reaction, living immature oocytes and eggs matured with 1-MA in seawater at various pH conditions were incubated with 25 µM of the synthetic fluorescent polyamine BPA-C8-Cy5 to label the extracellular matrix (File S1). The distinct excitation and emission spectra of BPA-C8-Cy5 and Alexa568-phallodin enabled us to label the extracellular matrix and F-actin simultaneously. The fluorescent polyamine BPA-C8-Cy5 also visualized the VL, and the sperm acrosomal processes during insemination in acidic or alkaline seawater of immature oocytes and mature eggs. Alexa568-phalloidin, on the other hand, allowed us to monitor the sequential changes of the cortical F-actin in the same oocyte/egg. The acrosomal processes in *A. aranciacus* sperm were visualized by diluting sperm in seawater at different pH containing the fluorescent polyamine before insemination in given conditions. The signals of the fluorescent probes from the samples were detected with a Leica TCS SP8X confocal laser scanning microscope equipped with a white light laser and hybrid detectors using the Lightning deconvolution mode (Leica Microsystem, Wetzlar, Germany). The number of eggs examined for each condition (n) is specified in the Results.

### 2.4. Visualization of Sperm Inside the Fertilized Eggs

Diluted sperm were stained with 5 µM Hoechst-33342 (Sigma–Aldrich, Saint Louis, MO, USA) for 30 s before insemination. The labeled sperm nuclei incorporated into *A. aranciacus* oocytes and eggs were counted in the cytoplasm of fertilized eggs 10–15 min after insemination using a cooled CCD (Charge-Coupled Device) camera (MicroMax, Princeton Instruments Inc., Trenton, NJ, USA) mounted on a Zeiss Axiovert 200 microscope with a Plan-Neofluar 40X/0.75 objective with a UV fluorescent lamp. The number of fertilized eggs examined (n) for each condition in three or four independent experiments (N) using three different females is shown in the Results and Table 1.

### 2.5. Statistical Analysis

The numerical MetaMorph data were compiled and analyzed with Excel (Microsoft Office 2010) and reported as mean ± standard deviation (SD) in all cases in this manuscript. Oneway ANOVA was performed through Prism 5.00 (GraphPad Software), and *p* < 0.05 was considered statistically significant. For ANOVA results showing *p* < 0.05, the statistical significance of the difference between the two groups was assessed by Tukey’s post hoc tests. The two groups of data showing significant differences from each other were marked with symbols indicating the p values in the figure legends. For each experimental condition, 5 or 6 oocytes were used, and they were from the same animal. Now, of course, the same experiment was repeated 2–4 times (N) utilizing the oocytes from as many (N) animals. Thus, N refers to the number of replicates. On the other hand, the numerical data for Ca^2+^ signal and other values were pooled together for statistical Analysis by using all the oocytes treated in the same way. This number of oocytes was referred to with ‘n’.

## 3. Results

### 3.1. Structural Changes of Cortical F-Actin in Immature Starfish Oocytes Incubated at pH 6.8 and pH 9

In the starfish *Astropecten aranciacus*, immature oocytes isolated from the ovary in natural seawater (NSW, pH 8.1) are arrested at the prophase of meiosis I. At this stage of the maturation process, they are penetrated by multiple sperm upon insemination (Appendix A) and fail to undergo the cortical changes, such as exocytosis of the cortical granules and separation of the vitelline layer (VL) from the egg plasma membrane that are experienced by the mature eggs. Indeed, when the immature oocytes are stimulated in vitro with 1-MA and inseminated, the optimum period to achieve a successful monospermic fertilization response is between the germinal vesicle breakdown (GVBD) and the formation of the first polar body. A fully grown *A. aranciacus* immature oocyte isolated from the ovary contains the large nucleus (GV) at the animal pole. The oocyte is surrounded by the jelly coat (JC) to which follicle cells (FC) adhere (Figure 1A). The microinjection of the fluorescent F-actin filament probe AlexaFluor 568-phalloidin in living immature oocytes (n = 18) allows the visualization of the network of actin filaments characteristic of the cytoplasm of the GV-stage oocytes. The VL and the JC of immature oocytes were visualized with the fluorescent polyamine BPA-C8-Cy5 (Figure 1B, green color). The addition of sperm double-labeled with the polyamine probe and the DNA dye Hoechst-33342 (Figure 1C) to the immature oocytes in NSW (pH 8.1) induced the formation of numerous blebs on the oocyte surface due to polyspermic insemination (Figure 1D, arrow and Figure 1E). The binding and fusion between the two gametes promote the formation of the fertilization cones due to the protrusion of the ectoplasmic region of the oocyte as the result of the reorganization of the cortical F-actin in preparation for sperm incorporation (Figure 1E, arrow and Figure 1F, arrowheads). Figure 1D also shows no separation of the VL from the oocyte surface after insemination due to the F-actin-dependent structural organization of the oocyte cortex, which is distinct from that of mature eggs concerning the microvilli length and the orientation of cortical granules [19,20,21].

Although the incubation of *A. aranciacus* immature oocytes in NSW at pH 6.8 for 20 min apparently did not remove the JC surrounding the oocytes (Figure 2C), it reduced blebs formation and sperm penetration following insemination of the oocytes in acidic seawater (Appendix A). The effect of the acidic seawater on the structure of the cortical actin cytoskeleton was evident in these oocytes (n = 12), which showed a clear difference in the F-actin fibers in the specific cortical regions that are formed to incorporate the sperm after insemination in acidic seawater (Figure 2C, arrowheads) as compared to those in the control oocytes shown in Figure 1F arrowheads. Furthermore, the transmission electron microscopy (TEM) micrograph in Figure 2E showed the presence of cortical granules (CG) in the cytoplasm of oocytes incubated in acidic seawater that were dislodged from the plasma membrane as a result of the structural rearrangement of the cortical F-actin. Upon insemination, occasionally, it was possible to observe sporadic exocytotic events on the cortex of the oocyte that did not expand to the entire oocyte surface (Figure 2E, arrow).

At variance with the low rate of sperm penetration in immature oocytes approached by sperm in acidic seawater, GV-stage oocytes incubated for 20 min and inseminated in NSW (pH 9) were penetrated by a more significant number of sperm as a result of the increased receptivity to the sperm, which is ascribable to the pH-dependent structural changes of the oocyte cortex (Appendix A). That the incubation of immature oocytes in alkaline seawater promoted the alteration of the cortical F-actin structure is also indicated by the elevation of the VL following exocytosis of the CG upon fusion and the subsequent release of their contents into the perivitelline space (n = 10). The confocal image (Figure 2D) showed numerous F-actin structures beneath the fertilization cones formed in the cortical cytoplasm to incorporate multiple sperm (arrowheads). However, it is worth noting that the elevation of the VL upon insemination of immature GV-stage oocytes in alkaline seawater, which was also visualized at TEM (Figure 2F), did not resemble that occurring in mature eggs (see Figure 3F for comparison) as it collapsed within a few minutes on the oocyte surface, unlike in the mature eggs. Similarly, the elevation of the VL was also inhibited when immature oocytes were microinjected with a fluorescent phalloidin and inseminated in seawater containing the fluorescent polyamine to visualize the formation of the spikes in the perivitelline space due to the F-actin extension with the confocal microscope.

### 3.2. Effect of the Acidic and Alkaline Seawater on the F-Actin-Dependent Maturation Process Induced in Vitro by the Hormone 1-MA

At variance with immature oocytes that permit entry of multiple sperm, mature eggs obtained by incubation of the same oocytes in the presence of 1-MA exhibit a normal fertilization response with monospermy due to the restructuration of the oocyte cortex occurring a few minutes after hormonal stimulation and around the time of GVBD [11,12,33]. Figure 3A shows an *A. aranciacus* egg matured with 1-MA for 70 min which is the time for this species to achieve the optimal physiological conditions necessary to respond correctly to the fertilizing sperm. The hormonal stimulation induces the disassembly of the nuclear envelope of the GV and the intermixing of the nuclear components with the cytoplasm [11]. Insemination in NSW (pH 8.1) activates the eggs that respond with the elevation of the fertilization envelope (FE) as a result of CG exocytosis (Figure 3B and Appendix A). Figure 3C,D shows a confocal image of the dramatic rearrangement of the F-actin on the outer region of the egg cytoplasm at fertilization. Before fertilization, the subplasmalemmal actin filaments were regularly oriented perpendicularly to the egg surface (Figure 3C, orange color), but appeared sparse after fertilization (n = 13). Figure 3D shows the network of actin filaments characteristic of the outer region of the cytoplasm of living mature eggs, surrounded by the VL and the JC (visualized in green color with BPA-C8-Cy5) in the same living egg. The image also shows the translocation of the actin filaments from the cortex of the fertilized egg to the center (orange color) and the elevation of the fertilization envelope (FE) and F-actin spike formation in the perivitelline space (PS), labeled with BPA-C8-Cy5 upon incubation of unfertilized eggs. The Hoechst-33342-stained sperm inside the cortical cytoplasm, visualized 6 min after insemination, is visible in blue beneath the fertilization cone (arrow). Transmission electron microscopy (TEM) observations revealed the ultrastructure of the surface of an unfertilized control egg matured with 1-MA for 70 min (Figure 3E). Microvilli (MV) distributed uniformly and regularly on the egg surface are embedded in the VL, covering the plasma membrane. Upon insemination, the VL, which will form the FE, separates from the egg plasma membrane following the release of the content of the CG in the PS (Figure 3F, arrow). Elongation of microvilli (MV) in the PS is also visible in the eggs fixed 5 min after insemination.

When GV-stage oocytes (*A. aranciacus*) are stimulated with 1-MA to induce maturation in acidic seawater (pH 6.8), the transmitted light microscopic images of GVBD and the elevation of the fertilization envelope (Appendix A) resembled those of the control eggs matured and inseminated in NSW (pH 8.1) shown in Figure 3A,B. However, subtle structural differences were observed in the cortex of mature eggs before and after fertilization due to the effect of the acidic seawater on the cortical F-actin. The confocal image of Figure 4A shows 1-MA-induced reorganization of the F-actin in the egg cortex (n = 14), highlighted by the conspicuous absence of F-actin filaments (orange color) that are typically oriented perpendicularly to the plasma membrane in control eggs, as exemplified in Figure 3C. Following fertilization and elevation of FE (green color in Figure 4B and the TEM micrograph in Figure 4D), the lack of the centripetal translocation of the actin fibers from the egg surface is evident in the confocal image of Figure 4B (orange color), with an F-actin remodeling events occurring in sea urchin eggs, too [42,45]. The effect of the acidic seawater on the microvillar morphology of the unfertilized mature egg is highlighted in the TEM micrograph (Figure 4C), showing MV of different lengths embedded in the VL and CG detached from the egg surface. Upon insemination, CG are no longer visible in the cortical cytoplasm due to their exocytosis, which is also evidenced by the released material (arrow) in the PS in the TEM image in D.

In immature oocytes stimulated with 1-MA for 70 min in alkaline seawater (pH 9), GVBD did not occur, as shown in the transmitted light image (Figure 5A and Appendix A). The confocal microscopic image of the cortical and cytoplasmic F-actin in the eggs matured with 1-MA in alkaline seawater (n = 13) shows, at variance with the control, the inhibition of the dramatic reorganization of the F-actin in the cytoplasm and cortex of the oocytes following GVBD and lack of the disappearance of the network of the actin filaments in the cytoplasm (Figure 5C, orange color). The outer region of the cytoplasm of the oocytes treated with 1-MA in alkaline seawater was morphologically altered, as shown by the ultrastructural analysis by TEM (Figure 5E). The image highlights an increase and not a decrease in microvillar length, the latter of which (shortening of microvilli) is a morphological change that usually occurs at the time of the GVBD and is thought to be essential for the establishment of a normal fertilization potential and Ca^2+^ response at fertilization [8,20,22,27]. Furthermore, the alkaline seawater induced an increased polyspermic incorporation rate, as judged by the higher number of F-actin structures formed to incorporate sperm into the cytoplasm (Figure 5D, arrowhead). The TEM micrograph (Figure 5F) of an egg matured in alkaline seawater, fertilized, and fixed 5 min upon insemination shows the elevation of the FE as a result of the exocytosis of the CG into the PS, which was not detectable under a transmitted light (B) and a confocal microscope (D) as it slightly elevated from the egg surface.

Since GVBD did not take place in the oocytes stimulated with 1-MA in seawater at pH 9, the GV-stage oocytes were first treated for 50 min with 1-MA in NSW (pH 8.1) to allow the breakdown of the nuclear envelope and then transferred to seawater at pH 9 for 20 min (Appendix A). A maturation time of 70 min is required to achieve optimal fertilizability conditions for the eggs of this starfish species. In these experimental conditions, even if the intermixing of the nucleoplasm with the cytoplasm normally occurred, 20 min of exposure to alkaline seawater (pH 9) compromised the fertilization response, as shown in Figure 6. The visualization of sperm entry in these eggs showed that, even if sperm incorporation was not affected (see below), the time required for the eggs to incorporate the sperm was much longer than in the control eggs fertilized in NSW (Figure 3B). Indeed, 18 min after insemination, the sperm is still trapped in the fertilization cone, as shown in Figure 6B (arrowhead). This is probably due to the alteration (stabilization) of the F-actin core of microvilli (MV), which was more prominent in these experimental conditions than in control (see the individual MV marked in the TEM image of Figure 6C). The alteration of the morpho-functionality of the F-actin of the egg cortex caused by the alkaline seawater-induced maturation process was also evident following fertilization. Figure 6B shows an accumulation of longer actin fibers (orange color) on one side of the egg surface 18 min after insemination that failed to translocate toward the center of the fertilized egg (n = 11), which the control eggs in NSW (pH 8.1) achieved in 6 min (Figure 3D). Nonetheless, the stabilization of the actin filaments in the egg cortex did not interfere with the elevation of the FE due to the exocytosis of CG in the PS (arrow), as shown in the confocal and TEM images in B and D.

### 3.3. Effect of the Acidic or Alkaline Seawater on the Polyspermy Rate of Immature Oocytes and the Sperm Entry Following Oocytes Maturation in Altered Seawater pH

To test whether the lower or higher pH of seawater could affect the polyspermy, characteristic of immature starfish oocytes, GV-stage oocytes were incubated for 20 min in NSW (pH 8.1, control), 6.8 (acidic), or 9.0 (alkaline) and subsequently inseminated with Hoechst 33342-stained sperm (Table 1). The number of oocyte-incorporated sperm was counted 10–15 min after insemination. Figure 7A and the histogram (green color) show the fluorescently labeled DNA of multiple sperm inside the oocytes inseminated in NSW at pH 8.1 (4.13 ± 0.90, n = 80). At variance with that, immature oocytes incubated and inseminated in seawater at pH 6.8 (Figure 7A, second fluorescent image in the panel and the brown color of the histogram) were penetrated by a lower number of sperm than the control (1.22 ± 0.75, n = 80, *p* < 0.01). Since the organization of the F-actin structures in the oocyte cortex that are formed to incorporate sperm is quite different in the oocytes inseminated in acidic seawater (Figure 2C), as compared to those in control (pH 8.1) (Figure 1F), the reduced rate of polyspermy observed in the oocytes inseminated in acidic seawater may be attributed to the alteration of the structural organization of the cortical F-actin. By contrast, GV-stage oocytes incubated and inseminated in alkaline seawater (pH 9) were penetrated by much more sperm than the control (15.35 ± 2.65, n = 80, *p* < 0.01), as shown in the fluorescent image of Figure 7A (third fluorescent image of the panel) and the histogram (purple color).

In a slightly different protocol, we examined the effect of acidic or alkaline seawater on oocyte maturation and its consequence at fertilization (Figure 7B and Table 1). As expected, *A. aranciacus* oocytes incubated with 1-MA for 70 min in NSW (pH 8.1) led to monospermic fertilization in most cases (48 out of 60 eggs). In the remaining mature eggs, the number of sperm counted was two or three per egg to make the average 1.23 ± 0.25 (n = 60). When oocytes were induced to undergo maturation with 1-MA in acidic seawater (pH 6.8) (Figure 7B and the histogram brown color), only 6 out of 60 mature eggs showed a single sperm in the cytoplasm. When GV-stage oocytes were induced to undergo maturation with 1-MA in NSW (pH 8.1) for 50 min and then transferred to seawater at pH 9 for further (20 min) incubation, the eggs experiencing the later stage of maturation in alkaline seawater exhibited a higher rate of polyspermy (Figure 7B, histogram labeled pH 8.1 + pH 9) than the control eggs. Of 60 mature eggs being analyzed under a fluorescence microscope, 34 eggs were monospermic, and the remaining eggs had 2 or 3 sperm inside to make the average 1.82 ± 0.46 sperm per egg (n = 60), which was modestly higher than the value in the control (non-significant). By contrast, when maturation was promoted in alkaline seawater (pH 9), a higher number of sperm entered the egg under these experimental conditions (20.42 ± 2.55 n = 60, *p* < 0.01 in comparison with the control NSW pH 8.1), which is also indicated by the visualization of one fertilized egg with seven sperm inside (Figure 7B).

### 3.4. Insemination of the GV-Stage Oocytes in Seawater with Altered pH Significantly Affects the Sperm-Induced Ca^2+^ Response

Previous studies from our laboratory showed that the sperm-induced Ca^2+^ response in the GV-stage starfish oocytes (not treated with the maturing hormone) is dependent on the structural organization of the oocyte cortex. At this stage of the maturation process (prophase I), the immature oocytes have longer microvilli [20,21] and more irregular actin meshwork underneath the plasma membrane in comparison with the mature eggs (Figure 1B and Figure 3C). The F-actin morphological features of the cortex of immature oocytes mirror their characteristic Ca^2+^ response at fertilization, which is quite different from that in mature and overmatured eggs [8].

Figure 8A shows that the addition of sperm to a GV-stage oocyte suspended in NSW (pH 8.1) promotes two small releases of Ca^2+^ due to the interaction of two sperm with the oocyte plasma membrane (arrowheads). These Ca^2+^ releases are followed by a subitaneous increase in Ca^2+^ at the periphery of the oocyte (cortical flash, CF) that lasts only a few seconds and is followed by two Ca^2+^ waves that converge and propagate to the opposite pole in about two minutes, as indicated by the pseudocolor image of the relative fluorescence analysis (n = 13). The graph in Figure 8D and the histogram in E (green colors) show that when the immature oocytes were inseminated in NSW, the time lag until the first Ca^2+^ release was 22.5 ± 7.7 s. The Ca^2+^ waves reached an amplitude of 0.37 ± 0.04 RFU in 358.7 ± 36.3 s. At variance with the control, oocytes inseminated in acidic seawater (pH 6.8) showed a Ca^2+^ response only in 4 out of 13 oocytes with a much longer delay (215.3 ± 87.5 s, *n* = 4, *p* < 0.01). The peak amplitude of the Ca^2+^ wave was also significantly lower than in the control (Figure 8D and E, blue colors). Moreover, the occurrence of the CF, which followed the Ca^2+^ waves elicited by two sperm visualized in the third pseudocolor image was substantially delayed as it appeared 1 min and 32 s after the initiation of the first Ca^2+^ signal, as shown in the fourth relative fluorescence image of Figure 8B. Differences were also observed in the time required to reach the highest release of Ca^2+^ (time to reach the Ca^2+^ peak, 0.19 ± 0.06 RFU in 494.7 ± 49.9 s, n = 4) as compared to the control (358.7 ± 36.3 s n = 13, *p* < 0.05) (see also the histogram of Figure 8E, blue color). As for the Ca^2+^ response induced by sperm addition to immature oocytes incubated in seawater at pH 9 (n = 14), even if the sequence of occurrence of the initial Ca^2+^ spots and CF was similar to that of the control, striking differences were observed in the pattern of Ca^2+^ release and the decline as shown in the pseudocolor images (Figure 8C–E). The results show that the time to reach the Ca^2+^ peak amplitude was faster than the control (167.6 ± 25.3 versus 358.7 ± 36.3 s, *p* < 0.01) and that the higher Ca^2+^ increase (0.45 ± 0.04 RFU, *p* < 0.05) induced by the fusion of multiple Ca^2+^ waves (Figure 8C, arrowheads) propagated faster than the control (44 s versus 120 s). The results also show that insemination of the oocytes kept in alkaline seawater dramatically alters the pattern of Ca^2+^ release by the heavy stimulation of the oocyte surface caused by the interaction and subsequent penetration of many more sperm (15.35 ± 2.65, n = 80) than in the control.

### 3.5. A. aranciacus Oocytes Matured in Seawater at pH 6.8 and pH 9 Show an Altered Ca^2+^ Response at Fertilization as Compared to the Control

In normal conditions, upon insemination, eggs matured in NSW (pH 8.1) with 1-MA at a final concentration of 10 µM experience the first Ca^2+^ response (the CF), which takes place simultaneously at the periphery of the egg cortex as a result of the activation of the L-type Ca^2+^ channels promoting Ca^2+^ influx [35]. The pseudocolor images show that the CF is followed by a Ca^2+^ wave propagating to the opposite pole (Figure 9A). The graph and histogram in D and E (green colors) show the fertilization Ca^2+^ responses representative of 13 control eggs matured and fertilized in NSW. The Ca^2+^ wave in control eggs reached a peak amplitude of approximately 0.49 ± 0.03 RFU and took about 2 min to get to the opposite pole (traverse time, 131.1 ± 16.1 s) (Figure 9A,E).

In addition to a normal Ca^2+^ response, optimal physiological conditions of the eggs are also required for a typical separation of the vitelline layer, which will form the FE seen in Figure 3B. The GV-stage oocytes stimulated with the hormone while suspended in acidic seawater (pH 6.8) can undergo GVBD and, if fertilized, display apparently normal elevation of the FE similar to that of the control. However, it is interesting to note that, at variance with immature oocytes in which incubation and insemination in seawater at pH 6.8 heavily reduced the Ca^2+^ response and the number of sperm entries, all the eggs matured in acidic seawater (n = 13) responded to sperm stimulation by elevating their intracellular Ca^2+^ with a CF, which was followed by a Ca^2+^ wave (Figure 9B). The Analysis of the Ca^2+^ changes following sperm addition showed that the only parameter of the sperm-induced Ca^2+^ response that was altered was the propagation time of the Ca^2+^ signal in the egg (Figure 9E), which was significant for a longer duration (160.7 ± 14.5 s).

On the other hand, when GV-stage oocytes were induced to mature in seawater at pH 9, they all failed to undergo GVBD. As aforementioned, 70 min after the addition of 1-MA, a GV with an abnormally elliptical shape was still visible (Figure 5A). Upon insemination, these “mature” eggs (n = 14) elicited a Ca^2+^ response that was heavily altered as compared to that of the control eggs matured and fertilized in NSW (Figure 9). The relative fluorescence images of Figure 9C show that at variance with the control, the Ca^2+^ response could initiate with a Ca^2+^ spot (arrowhead) which was then followed by a CF and a series of Ca^2+^ waves as a result of the fusion of multiple sperm transducing several Ca^2+^ signals in 8 eggs out of 14. The Ca^2+^ waves then converged and propagated faster to the opposite pole (101.4 ± 11.6 s, n = 14) than the control (131.1 ± 16.1 s, n = 13, *p* < 0.05). The analysis of the relative fluorescence changes has also indicated that 3 out of 14 eggs did not elicit any CF. Furthermore, the Ca^2+^ peak amplitude was also affected when eggs were matured in alkaline seawater (0.35 ± 0.04 RFU, *p* < 0.01), as shown in the graph of Figure 9D (brown colors) and the histogram in Figure 9E. A lesser effect on the Ca^2+^ response was highlighted in eggs inseminated after being matured for 50 min in NSW (pH 8.1) and 20 min in seawater at pH 9 (n = 12). While the Ca^2+^ wave always followed the CF as in control in A, its amplitude was significantly lower than that of the control (0.37 ± 0.02 RFU, *p* < 0.01), as shown in the graph in D (light brown color) and histogram in Figure 9E.

## 4. Discussion

Eggs of marine invertebrates have been broadly used in laboratories to study fertilization because the eggs are spawned freely into the sea, where they are immediately mixed with the spermatozoa. Thus, the fertilization process and the following developmental changes of the embryo can be easily observed in the petri dish. It has been known that the structural organization of the cortical F-actin in the oocytes and eggs profoundly impacts how they respond to the fertilizing sperm with Ca^2+^ increase and sperm incorporation [8,28,34,35,42]. Interestingly, our recent studies with sea urchin eggs in acidic and alkaline seawater conditions suggested that the pH of the seawater has profound effects on many aspects of fertilization that are linked to the cortical F-actin structural dynamics, such as patterns of Ca^2+^ signaling and actin cytoskeletal remodeling that take place in the eggs after fertilization [39,46].

In this communication, by using starfish as the experimental model, we have surveyed the effect of acidic and alkaline seawater not only on fertilization, but also on meiotic maturation. To begin with, the structural organization of the GV-stage oocytes and mature eggs are strikingly different in terms of the actin cytoskeleton (including microvilli) and the distribution of cortical granules and vesicles [37,40]. Because of that, immature oocyte insemination results in polyspermic entries. However, various manipulation of the pH of the media (seawater) either intensified the tendency of polyspermy or alleviated it (Table 1). In line with that, the changes of the pH of the incubation media in various conditions resulted in remarkable alterations in terms of organization of the F-actin cortical network, meiotic progression (GVBD), Ca^2+^ signaling pattern upon insemination, sperm incorporation, and cortical granule exocytosis in some cases. The results are summarized in Table 2. Another advantage of using starfish is that the formation of the acrosomal process at the head of sperm is much more prominent in starfish than in sea urchins. Furthermore, the location of the occurrence of the sperm acrosomal process is well known. Indeed, both in vitro and in vivo, it has been shown that the acrosomal process (AP) formation occurs when the fertilizing sperm contacts the outer layer of the jelly coat [26,29,43]. The tip of the long (20 µm) and thin acrosomal process made by actin filaments enters openings in the vitelline layer [21] and interacts with the egg plasma membrane to trigger electrical and Ca^2+^ changes upon their fusion [27,28].

Immature oocytes isolated from the gonad of *A. aranciacus* in natural seawater (pH 8.1) are prone to polyspermy due to their structural organization of the oocyte surface and cortex [7,8,20,21]. Following insemination, the interaction of the tips of the sperm acrosomal filament with the oocyte plasma membrane elicits numerous Ca^2+^ responses with a pattern of Ca^2+^ release (Figure 1B and Figure 8A), which is different from the single one produced by a mature egg undergoing regular cortical F-actin restructuring during maturation (Figure 3C and Figure 9A). The different organization of the cortical F-actin in the female gametes at the two different maturation stages is also indicated by the lack of the separation of the vitelline layer from the oocyte plasma membrane at fertilization (Figure 1D and Figure 3B). Finally, the incorporation of numerous sperm by the specialized F-actin structures polymerized in the oocyte cytoplasm beneath the fertilization cones (Figure 1F, arrowheads) further shows differences in the F-actin-dependent mode of sperm incorporation in comparison with mature eggs [8,21].

In starfish oocytes, following hormonal stimulation, the cortical cytoskeletal reorganization by G-proteins’ activation modulates the Ca^2+^ signals during the early and late phases of the maturation process in starfish [10,11,33,47,48]. Acidic and alkaline seawater incubation also altered the structure of the F-actin of the immature oocyte cortex, which, in turn, impaired the sperm-oocyte binding and the pattern of the sperm-induced Ca^2+^ signals. The role of intracellular pH in regulating actin polymerization and Ca^2+^ signaling in eggs has been demonstrated in sea urchins and *Xenopus* [49,50]. In this regard, besides the faster and higher Ca^2+^ release upon insemination in alkaline seawater, immature oocytes displayed a faster decline in the intracellular Ca^2+^ level down to the baseline level (Figure 8D, brown lines and histograms in Figure 8E), as previously shown by sea urchin eggs [40], as well as an increase in the rate of F-actin dependent sperm incorporation (Figure 7A and Table 1).

An additional indication of the striking effect of the higher pH on the structure of the cortical F-actin is given by the observation under a light microscope that the vitelline layer could separate from the plasma membrane of immature oocytes incubated and inseminated in alkaline seawater (Figure 2B). Since such an event never happens in control, these results add weight to the suggested role played by the state of polymerization of the F-actin of the cortex of immature oocytes and mature eggs in the regulation of the exocytosis of cortical granules and independently of the Ca^2+^ increase [10]. When maturation was induced in alkaline seawater, the inhibition of the structural changes of the cortical F-actin was also striking (see elongation instead of shortening of microvilli in Figure 5E). The cortical F-actin alteration also inhibits the breakdown of the nuclear envelope of the large nucleus GV anchored to the oocyte surface by actin filaments [23], the disassembly of which is dependent on the F-actin dynamic changes at the nucleoplasmic face that are essential to rupture the nuclear envelope [51,52,53]. As a result, the absence of the intermixing of the nucleoplasm with the cytoplasm by preventing the rearrangement of the cortical F-actin at the time of GVBD [11] impairs the acquisition of cytoplasmic maturity, a normal fertilization potential and Ca^2+^ response upon insemination of mature eggs [11,12,17,22,54,55].

Interestingly, the given condition of acidity (pH 6.8) did not make much difference to the morphology of the JC of immature oocytes, as judged by the fluorescent labeling by BPA-C8-Cy5 (compare Figure 1B and Figure 2C), and did not affect sperm motility as they arrived in the vicinity of the surface of the oocyte with no time lag. However, the pH appears to affect the sperm acrosomal process formation. As shown in Figure 2D, the acrosomal processes traversing the JC are more easily labeled in the oocytes inseminated in the alkaline seawater (pH 9). On the other hand, insemination in acidic conditions (pH 6.8) made it more challenging to observe the acrosomal processes over the oocytes compared with the oocytes inseminated in NSW pH 8.1 (compare Figure 1E and Figure 2C). This observation is comparable with the idea that the formation of the acrosomal process per se is facilitated in higher seawater pH. This is possibly due to the effect of the seawater pH on the sugar moiety of the sulfated glycoprotein, i.e., the inducer of the acrosome reaction [56].

Interestingly, at fertilization of oocytes matured in acidic seawater, only the initiation of the sperm-induced Ca^2+^ response was not affected, indicating a standard functionality of the JC but the propagation of the Ca^2+^ wave and sperm entry were compromised due to the alteration of the cortical F-actin dynamics (Figure 7B and Figure 9 and Table 1 and Table 2). Drastic increases in the Ca^2+^ response and in the rate of sperm penetration were observed in GV-stage oocytes challenged with sperm and after maturation of oocytes in alkaline seawater. In light of the fact that the acrosomal processes are more easily visible in alkaline seawater and much less so in acidic seawater (Figure 2C,D), this enhanced sperm entry into the oocytes and eggs is probably due to the facilitated induction of the sperm acrosome reaction in starfish, which is known to be stimulated by an increased pH [26,29]. Furthermore, the results showed that GVBD inhibition was due to the effect of the alkaline seawater on the F-actin structures of microvilli and cortex, which failed to undergo morphological changes underlying the maturation process (Figure 5A). Thus, alkaline seawater may not represent the optimal pH for cdk1 activity and for the F-actin rearrangement to fragment nuclear membranes leading to GVBD [3,4,51]. Finally, even if a less drastic effect of the alkaline seawater on polyspermic was observed when immature oocytes were stimulated with 1-MA for 50 min in NSW and 20 min in alkaline seawater, the time delay in sperm incorporation and alteration of the F-actin translocation following fertilization also indicated the strict interdependence of the effect of the alkaline pH on the structural modification of the cortical F-actin, leading to an altered Ca^2+^ response at fertilization (Figure 9).

The advantage of designing controlled experimental conditions in the laboratory to study the effect of the altered seawater pH on the physiology of the fertilization process of marine eggs is of great importance. Indeed understanding how changes in the pH values of seawater affect the structural organization and dynamics of the actin cytoskeleton of the egg cortex may shed light on the molecular mechanisms regulating the F-actin-dependent Ca^2+^ signaling in other cell types as well. In line with this, acidic and alkaline seawater alteration of the cortical actin cytoskeleton of starfish oocytes, which is differently organized in the two oocyte maturation stages (this contribution), and sea urchin eggs [37,39,40] affect the sperm-induced Ca^2+^ signaling and the subsequent F-actin reorganization necessary for cleavage [38,39]. These results may also light up a similar scenario occurring in natural environmental conditions, i.e., reducing the surface ocean pH due to the absorption of the increasing atmospheric carbon dioxide. Even if a seawater pH 6.8 would correspond to an extreme condition, seawater acidification will inevitably limit the animal population by significantly impacting the initiation of the fertilization process.

## 5. Conclusions

In starfish, it has been well known that the breakdown of the immature oocytes’ large nucleus (germinal vesicle, GV) induced by hormonal stimulation must occur to make the mature egg ready to be successfully fertilized with normal Ca^2+^ response and penetration by only one sperm. The oocyte maturation process following the addition of 1-MA brought about by the intermixing of the nucleoplasm with the cytoplasm ensures the F-actin-dependent reorganization of the cortex of the mature egg for a normal fertilization response. In line with this, immature oocytes are prone to polyspermic fertilization. The results of this contribution have shown that the acidic or alkaline seawater pH reduced or increased polyspermy rates, respectively, and the altered Ca^2+^ responses at fertilization. Altered pH also affects the acrosome reaction of the sperm and the structure and dynamics of the actin filaments in the oocyte cortex. Morphological modifications of the cortical F-actin network of eggs matured in seawater at different pHs influencing the sperm-induced fertilization response were also observed. The results have highlighted the strict interdependence between the seawater pH and the sperm-induced Ca^2+^ signals in immature oocytes and mature eggs, which reflect the different organization of the cortical F-actin in their two maturation stages.

## Figures and Tables

**Figure 1 cells-12-00740-f001:**
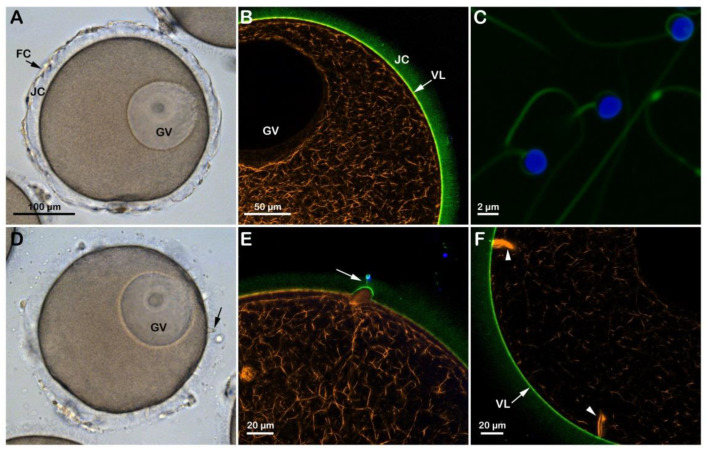
Structural changes of the actin cytoskeleton following insemination of immature oocytes in NSW (pH 8.1). (**A**) *A. aranciacus* oocytes isolated from the ovary in NSW (pH 8.1) are blocked at the prophase I of the meiotic division, showing a large nucleus (germinal vesicle, GV). These oocytes also surrounded by the follicle cells (FC) adhering to the extracellular jelly coat (JC). (**B**) In confocal microscopy, actin filaments were visualized with AlexaFluor 568-phalloidin (orange color), and the vitelline layer (VL) and JC were disclosed with fluorescent polyamine BPA-C8-Cy5 (green color). (**C**) Sperm heads were labeled with Hoechst-33342 (blue color) and the tails with BPA-C8-Cy5. (**D**) Upon sperm addition, several fertilization cones (i.e., polyspermic) are formed on the oocyte surface as a result of the protrusion of the cytoplasm (blebs, arrow). (**E**) JC triggers the formation of the acrosomal process, which is now visible as if there is a tether between the sperm head and the fertilization cone (arrow). (**F**) Formation of thick bundles of F-actin where multiple sperm are incorporated.

**Figure 2 cells-12-00740-f002:**
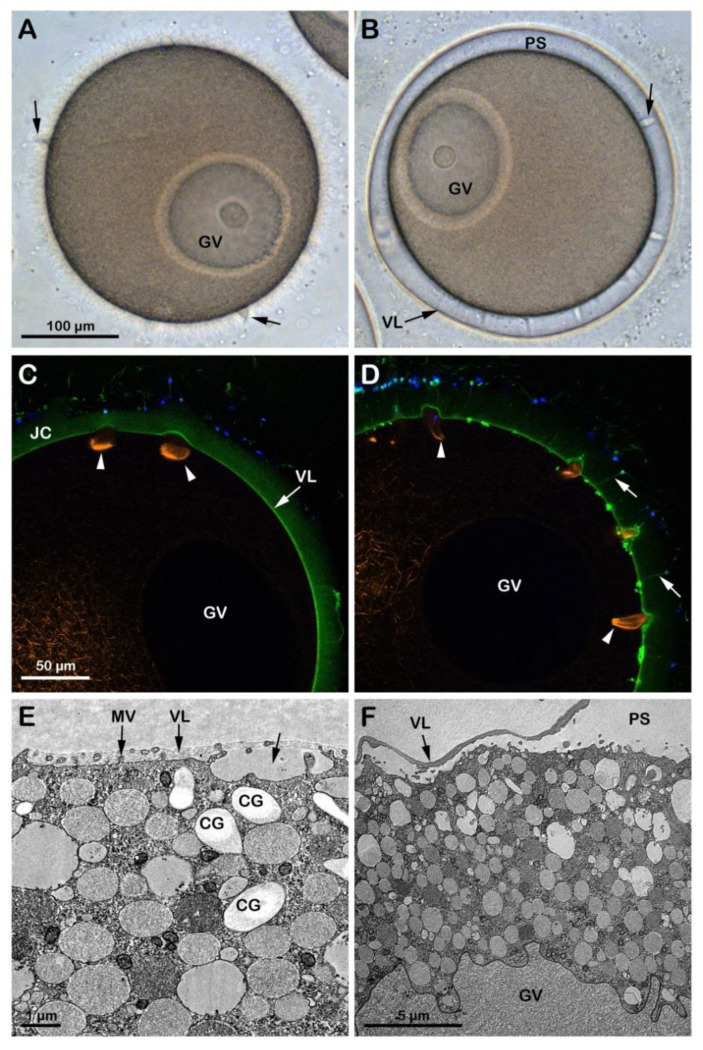
Penetration of sperm into GV-stage oocytes incubated and inseminated in acidic (pH 6.8, left panels) and alkaline (pH 9, right panels) seawater. (**A**) In the acidic seawater, the morphology of the fertilization cones (blebs, arrows) was similar to that of the oocytes fertilized in NSW at pH 8.1 (Figure 1A), and there was no elevation of the fertilization envelope (FE). (**B**) In alkaline seawater, FE elevates but the oocytes are predominantly polyspermic. (**C**,**D**) The shapes of the fertilizing cones labeled with AlexaFluor 568-phalloidin (orange color, arrowheads) are strikingly different in oocytes fertilized in acidic (round) and alkaline (sickle-formed) seawater. In alkaline seawater, a strikingly higher number of sperm entered the eggs, and the acrosomal processes were more easily visualized by BPA-C8-Cy5 (arrow) in the perivitelline space. (**E**) The TEM images show that cortical granules (CG) are dislodged from the surface of the oocytes in acidic seawater, which hampered their exocytosis and resulted in a lack of FE elevation. (**F**) In alkaline seawater, the VL readily elevates upon fertilization, creating a vast perivitelline space (PS). However, an even higher number of sperm entered (see the text). Note that microvilli (MV) in the oocytes of the acidic seawater (**E**) are not remodeled, unlike the ones in the alkaline seawater (**F**).

**Figure 3 cells-12-00740-f003:**
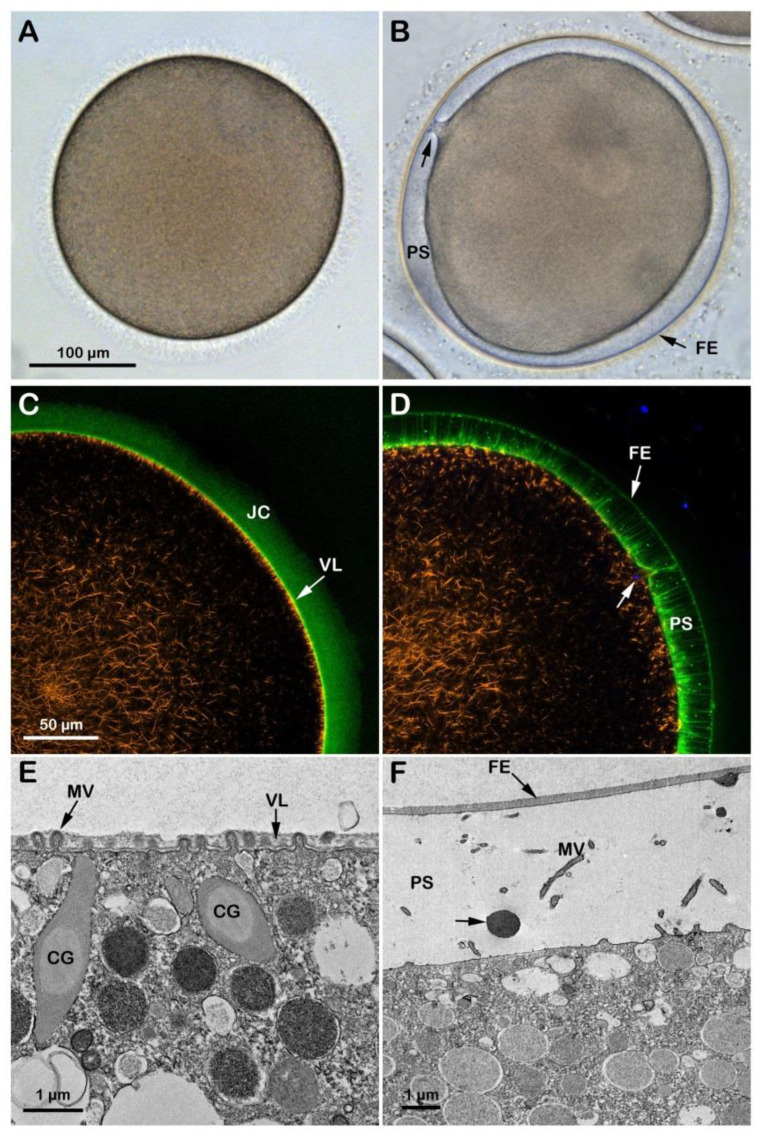
Cortical reaction and F-actin changes the eggs matured and fertilized in NSW pH8.1. GV-stage oocytes of *A. aranciacus* were stimulated with the maturing hormone 1-methyladenine (1-MA) in NSW (pH 8.1) for 70 min prior to fertilization. The left panels represent the eggs before fertilization, and the right ones 5 min after fertilization. (**A**) Transmission view of the mature egg. GV breakdown is evident. (**B**) Fertilized eggs show the elevation of the fertilization envelope (FE) and the expansion of the perivitelline space (PS). Arrow indicates the single site of sperm entry. (**C**,**D**) Confocal microscopy images showing actin filaments (AlexaFluor568-phalloidin, orange color), extracellular matrix jelly coat (JC and VL; BPA-C8-Cy5, green color), and sperm head (Hoechst-33342, blue color). The subplasmalemmal actin meshwork in an unfertilized egg (**C**) disappears as many microvilli are extended in the PS (**D**). (**E**,**F**) TEM images of the egg surface before (**E**) and after fertilization (**F**). By 5 min after fertilization (**F**), the content of the CG (arrow) is deposited in the perivitelline space (PS) containing elongated microvilli (MV).

**Figure 4 cells-12-00740-f004:**
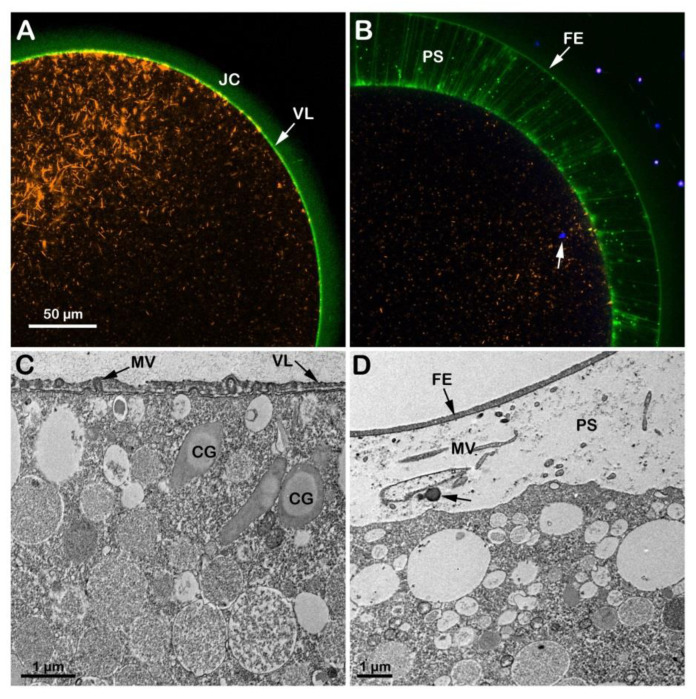
Structural changes of the actin cytoskeleton in the cortex of *A. aranciacus* eggs matured and fertilized in acidic seawater (pH 6.8). Left panels before fertilization, right ones after. (**A**,**B**) Confocal images. (**C**,**D**) TEM images. The elevation of the fertilization envelope (FE) following insemination still occurs in this acidic condition (**B**,**D**). Note that the jelly coat (JC) is still visible in the eggs matured in acidic seawater (panel (**A**); JC visualized by BPA-C8-Cy5, green color). The arrow in (**B**) shows the DNA-labeled sperm in the cytoplasm of the fertilized egg. Following oocytes maturation in acidic seawater, the impairment of the cortical F-actin remodeling is also evident in the TEM image in (**C**), showing microvilli (containing actin filaments) with a different length embedded in the vitelline layer (VL). The cortical granules (CG) are not positioned near the plasma membrane, unlike the eggs matured in NSW with pH 8.1. Microvilli (MV) elongation in the perivitelline space (PS) 5 min after insemination is visible in the TEM image in (**D**).

**Figure 5 cells-12-00740-f005:**
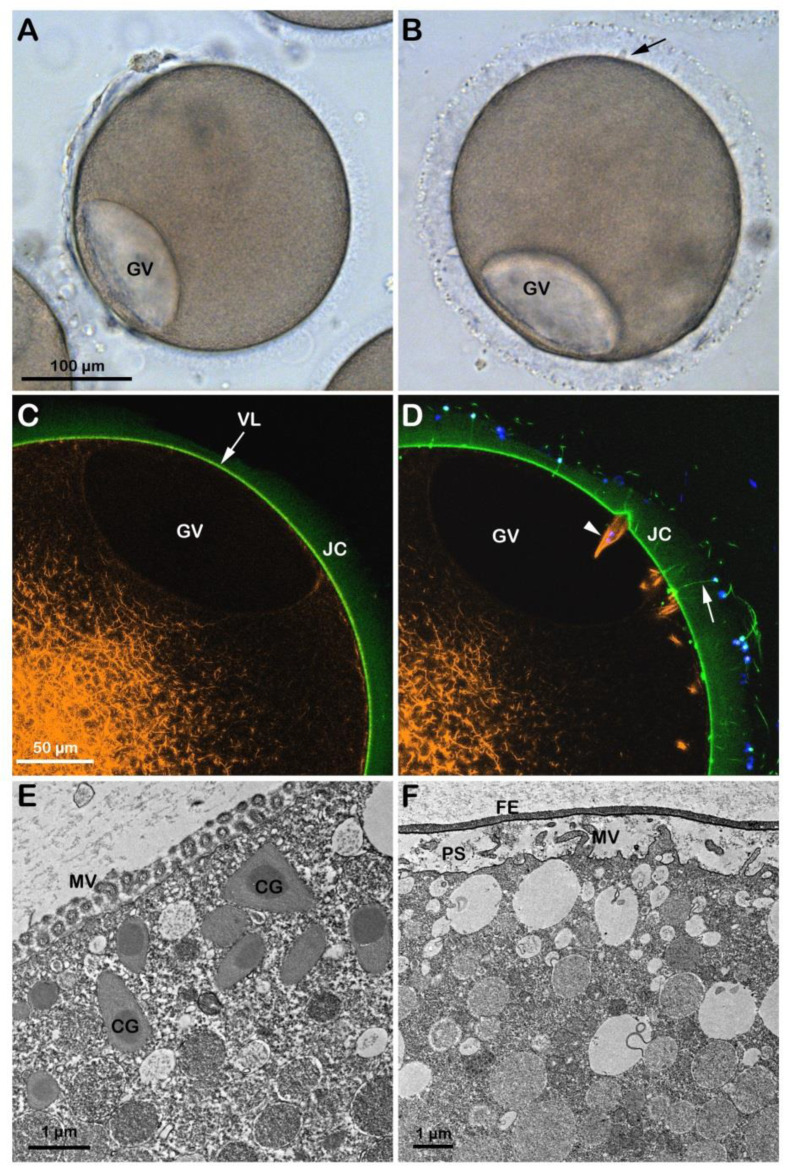
Starfish oocytes stimulated with maturation hormone in alkaline seawater fail to undergo GVBD. The GV-stage oocytes of *A. aranciacus* were exposed to the alkaline seawater (pH 9) for 70 min before insemination in the same media. The left panels represent images before insemination, and the right ones 5 min after insemination. (**A**,**B**) Light microscopy images. (**C**,**D**) Confocal micrographs. (**E**,**F**) TEM images. Note that incubation and maturation of GV-stage oocytes in alkaline seawater (pH 9) blocks the disassembly of the nuclear envelope of the GV (GVBD). However, the GV in these oocytes appeared flattened after the incubation in the alkaline seawater. Note also the formation of many fertilization cones containing F-actin structures apparently encompassing the head of incorporated sperm ((**D**), orange and blue colors, arrowhead). Numerous acrosomal filaments (arrow) impressively traversing the JC ((**D**), arrow; stained with BPA-C8-Cy5, green color), indicating high tendency of polyspermy. The elevation of the fertilization envelope (FE) 5 min after sperm addition is modest, which was visible only at the TEM level (**F**), and not at light microscopy (**B**). Microvilli (MV) elongate in the perivitelline space (PS) in (**F**).

**Figure 6 cells-12-00740-f006:**
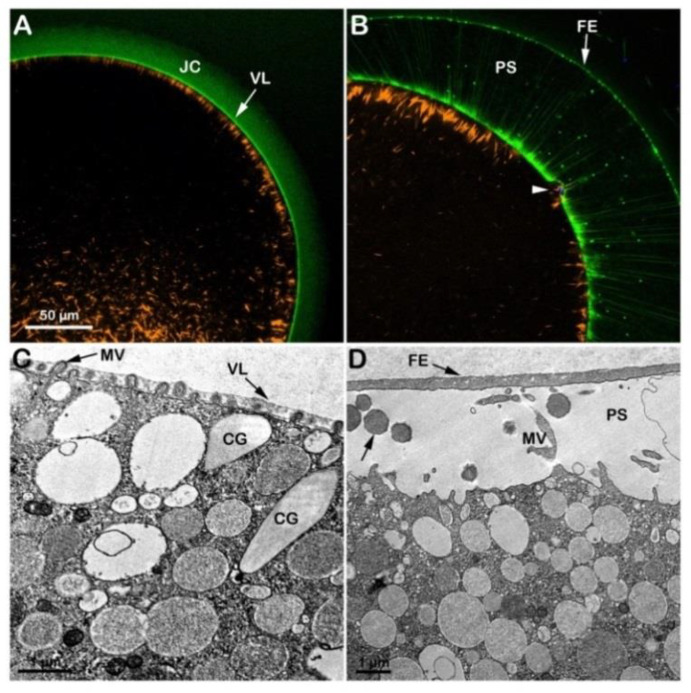
Cortical F-actin dynamics in *A. aranciacus* oocytes that were initially matured in NSW (pH 8.1) for 50 min and then in alkaline seawater (pH 9) for 20 min. The confocal (**A**,**B**) and TEM (**C**,**D**) were captured before (**A**,**C**) and after insemination (**B**,**D**). In the confocal images, BPA-C8-Cy5 (green color) delineated the vitelline layer (VL) and jelly coat (JC), and AlexaFluor 568-phalloidin (orange color) visualized F-actin. The confocal image in (**B**) shows that even if the elevation of the fertilization envelope (FE) is not affected, the translocation of the subplasmalemmal actin filaments (orange color) is much delayed. Note also the fertilizing sperm trapped beneath the fertilization cone in panel (**B**) (arrowhead) even 18 min after insemination. (**C**) After the second exposure to the alkaline seawater, the eggs exhibited prominent actin filaments within the core of some microvillar (MV, arrow), which were never observed in control eggs (Figure 3E). The TEM micrograph in panel (**D**) shows the cortical granules (arrow) content in the perivitelline space (PS) following their exocytosis.

**Figure 7 cells-12-00740-f007:**
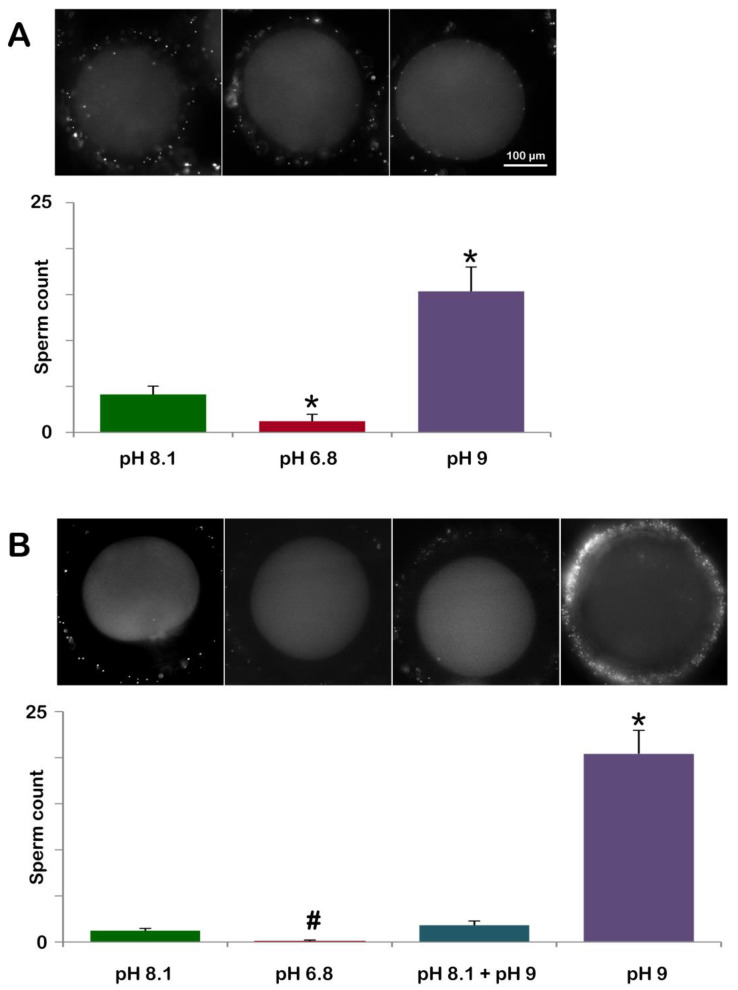
Pretreatment of GV-stage oocytes in acidic or alkaline seawater alters the rate of polyspermy at fertilization and sperm entry following oocyte maturation in altered seawater pH. (**A**) *A. aranciacus* immature oocytes were preincubated for 20 min in NSW at pH 8.1 (control), pH 6.8 (acidic), or pH 9 (alkaline) and inseminated in the same media with Hoechst 33822-prestained sperm. The number of oocyte-incorporated sperm 10-15 min after insemination was counted using epifluorescence microscopy for each case and presented in histograms. (**B**) Sperm incorporation in *A. aranciacus* eggs matured with 1-MA in a variety of pH conditions prior to fertilization: (i) NSW (pH 8.1) for 70 min; (ii) acidic seawater (pH 6.8) for 70 min; (iii) 50 min in NSW (pH 8.1) and then transferred to alkaline seawater (pH 9) for 20 min; (iv) alkaline seawater for 70 min. The number of sperm inside the eggs was counted in the same method and presented in the histograms. # *p* < 0.05, * *p* < 0.01.

**Figure 8 cells-12-00740-f008:**
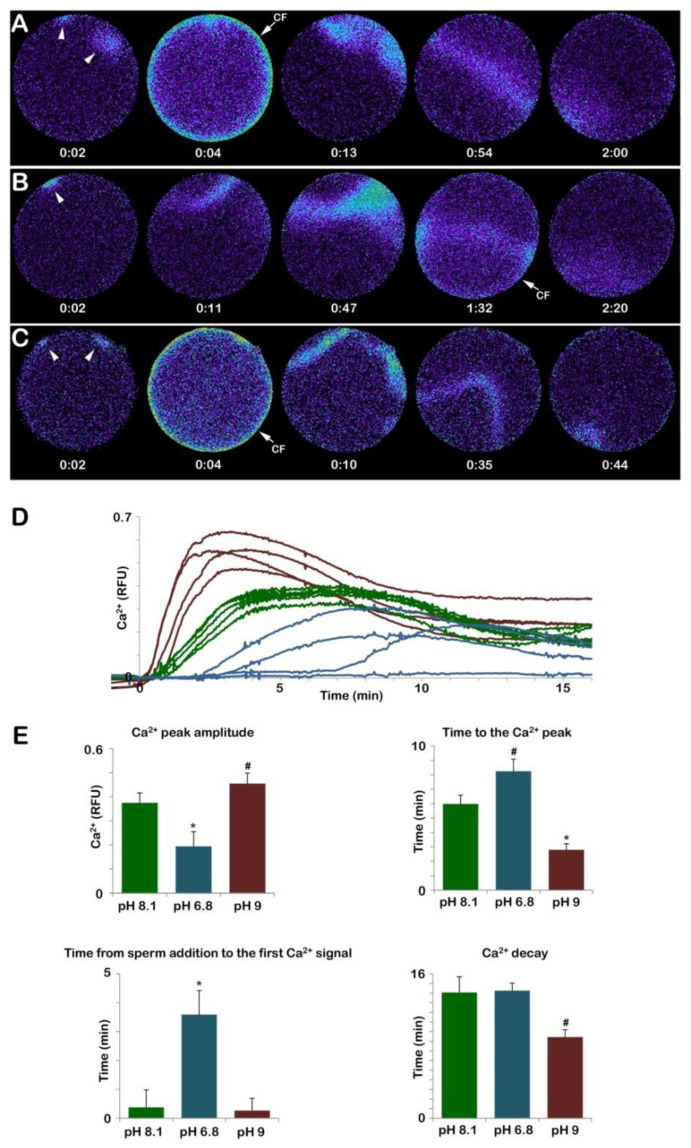
Insemination of GV-stage oocytes in acidic and alkaline seawater alters the sperm-induced Ca^2+^ response. The pseudocolored images in (**A**) represent the instantaneous increases in Ca^2+^ levels extracted from a time-lapse acquisition following the insemination of an immature *A. aranciacus* oocyte kept in normal seawater (NSW, pH 8.1). The first Ca^2+^ signal detected upon sperm addition is the initiation of multiple Ca^2+^ waves (arrowheads) as a result of a polyspermic stimulation that is followed by a cortical Ca^2+^ release (the cortical flash, CF), which occurs simultaneously at the periphery of the oocyte cortex. The Ca^2+^ waves run together to propagate to the opposite pole. Immature oocytes loaded with the calcium dye were fertilized in acidic (pH 6.8, (**B**)) and alkaline seawater (pH 9, (**C**)). The traces of Ca^2+^ signals were quantified in (**D**,**E**). Color codes: green, pH 8.1; blue, pH 6.8; purple, pH 9. Tukey’s post hoc test * *p* < 0.01, # *p* < 0.05. RFU = Relative Fluorescence Unit. The time in the pseudocolor fluorescent images (**A**–**C**) indicates minutes and seconds from the standard time format (mm: ss).

**Figure 9 cells-12-00740-f009:**
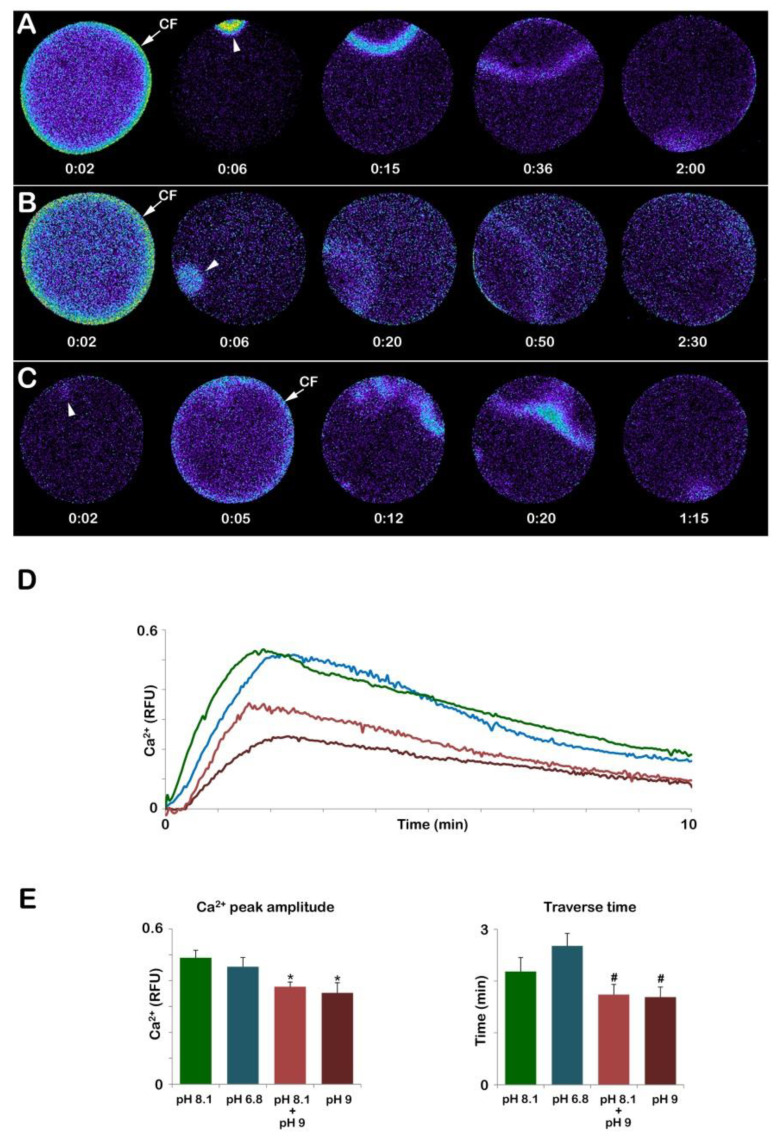
The Ca^2+^ response at fertilization mature eggs of *A. aranciacus* was altered when immature oocytes were induced to undergo maturation in acidic and alkaline seawater. (**A**) Ca^2+^ response in the control eggs matured and fertilized in NSW (pH 8.1). (**B**) Ca^2+^ response in the eggs matured and fertilized in the acidic seawater (pH 6.8). (**C**) Ca^2+^ response in the eggs matured and fertilized in the alkaline seawater (pH 9). The pseudo-color images of the Ca^2+^ response in the eggs first matured in NSW (pH 8.1) then fertilized in alkaline seawater sere similar to that the control eggs, and thus not shown here. (**D**,**E**) Quantification of the Ca^2+^ response in each condition. Note that in the eggs matured and fertilized in alkaline seawater (**C**), the cortical flash CF) precedes the Ca^2+^ wave, unlike the eggs in all other conditions, and that the generation of multiple Ca^2+^ signals in different areas indicate polyspermic gamete interaction. The higher pH of the seawater significantly reduced the peak amplitude of the Ca^2+^ wave (**D**,**E**), and shortened the propagation time required for the Ca^2+^ waves to reach the opposite pole. Tukey’s post hoc test * *p* < 0.01, # *p* < 0.05. RFU = Relative Fluorescence Unit.

**Table 1 cells-12-00740-t001:** Summary of the effect of the acidic or alkaline seawater on sperm entry in GV-stage oocytes and mature eggs. The letters (A to G) in the column of the experimental conditions marked with the asterisk refer to the experimental scheme specified in Appendix A.

Experimental Conditions *	n. of Inseminated Oocytes/Eggs	n. of Oocytes/Eggs Successfully Penetrated by Sperm	Percentage of Monospermy	Percentage of Polyspermy
A	80	80	0%	100%
B	80	38	15.8%	84.2%
C	80	80	0%	100%
D	60	60	80%	20%
E	60	6	100%	0%
F	60	60	0%	100%
G	60	60	56.7%	43.3%

**Table 2 cells-12-00740-t002:** Summary of the effect of the acidic or alkaline seawater on GV-stage oocytes and eggs matured in different seawater pH at fertilization. The letters indicating the experimental conditions refer to those provided in Appendix A.

Experimental Conditions	Morphology (Light and TEM Observations)	F-Actin Distribution Before and After Insemination	Ca^2+^ Changes	Sperm Incorporation	FE Elevation
A	Long MV CG dislodged from PM	Network of F-actin in the oocyte cytoplasm. Formation of fertilization cones after insemination.	CF after or together with multiple CW	Polyspermy	NO
B	Long MV CG dislodged from PM	Altered distribution of the cortical F-actin before insemination. Reduced formation of the fertilization cones.	Failure of sperm- induced Ca^2+^ release. Delay and reduced amplitude of CF and CW	Reduced polyspermy	NO
C	Long MV CG exocytosis at insemination	Altered F-actin redistribution. Increased formation of the fertilization cones.	Higher CW amplitude. Multiple CW Faster Ca^2+^ reuptake	Increased polyspermy	YES, but collapsed
D	Shortened MV CG beneath PM	F-actin fibers perpendicularly oriented in the unfertilized egg cortex. Centripetal translocation of F-actin fibers following fertilization. One fertilization cone.	CF before CW Single CW	Monospermy	YES
E	CG detached from egg surface Different length of MV	Lack of F-actin distribution in the unfertilized egg cortex following maturation. Inhibition of actin fibers translocation following fertilization.	Slower CW propagation	Inhibition of sperm entry	YES
F	Longer MV GVBD inhibition	Altered F-actin organization in the cortex and cytoplasm following maturation and fertilization. Increased number of fertilization cones.	CF after multiple CW Reduced CW amplitude Faster CW propagation	Polyspermy	NO
G	F-actin core of MV more evident	Alteration of the cortical F-actin organization following maturation and translocation upon fertilization.	Reduced CW amplitude	Monospermy	YES

Abbreviations: Microvilli (MV), Plasma Membrane (PM), Cortical Ca^2+^ Flash (CF), Ca^2+^ Wave (CW), Cortical Granules (CG), Fertilization Envelope (FE), Germinal Vesicle Breakdown (GVBD), Transmission Electron Microscopy (TEM).

## Data Availability

Not applicable.

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
