# Peer review of "The Effect of Acidic and Alkaline Seawater on the F-Actin-Dependent Ca2+ Signals Following Insemination of Immature Starfish Oocytes and Mature Eggs"

_cells, 2023, doi:10.3390/cells12050740_

Round 1
Reviewer 1 Report
The study is interesting, the results are of high quality and the different Figures very impressive.
In my opinion the manuscript needs a better organization.
The goal of the study should be clearly justified. Why are oocytes/eggs incubated in acid and alkaline sea water? Has it environmental relevance? On which base pH was selected?
The result section is long, it contains also literature citations (eg. lines 230-232). The legend of Figures is repetitive to the text and could be per se be sufficient as description of the results.
Some data are quantitative and might be reported in this way (elongation of microvilli)
Please report unit in Figures 8a-c. Is this the time scale in msec?
Figure 9 e: What means the combination of pH 8.1 + 9.0?
The authors could think to shorten the text of the results and summarize the major structural and physiological results in form of Table. This kind of presentation could give a comprehensive overview and allow a direct comparison of different experiments.
Are the spermatozoa affected by acid and alkaline pH (swimming velocity, acrosome reaction, type of swimming) and can this have an effect on fertilization processes?
Introduction: Lines 111-118 are a summary of results and should be deleted from introduction
Discussion: lines 634 – 658 might better be suited for introduction, lines 658-664 for Material and methods
The study is interesting, the results are of high quality and the different Figures very impressive.
In my opinion the manuscript needs a better organization.
The goal of the study should be clearly justified. Why are oocytes/eggs incubated in acid and alkaline sea water? Has it environmental relevance? On which base pH was selected?
The result section is long, it contains also literature citations (eg. lines 230-232). The legend of Figures is repetitive to the text and could be per se be sufficient as description of the results.
Some data are quantitative and might be reported in this way (elongation of microvilli)
Please report unit in Figures 8a-c. Is this the time scale in msec?
Figure 9 e: What means the combination of pH 8.1 + 9.0?
The authors could think to shorten the text of the results and summarize the major structural and physiological results in form of Table. This kind of presentation could give a comprehensive overview and allow a direct comparison of different experiments.
Are the spermatozoa affected by acid and alkaline pH (swimming velocity, acrosome reaction, type of swimming) and can this have an effect on fertilization processes?
Introduction: Lines 111-118 are a summary of results and should be deleted from introduction
Discussion: lines 634 – 658 might better be suited for introduction, lines 658-664 for Material and methods
The study is interesting, the results are of high quality and the different Figures very impressive.
In my opinion the manuscript needs a better organization.
The goal of the study should be clearly justified. Why are oocytes/eggs incubated in acid and alkaline sea water? Has it environmental relevance? On which base pH was selected?
The result section is long, it contains also literature citations (eg. lines 230-232). The legend of Figures is repetitive to the text and could be per se be sufficient as description of the results.
Some data are quantitative and might be reported in this way (elongation of microvilli)
Please report unit in Figures 8a-c. Is this the time scale in msec?
Figure 9 e: What means the combination of pH 8.1 + 9.0?
The authors could think to shorten the text of the results and summarize the major structural and physiological results in form of Table. This kind of presentation could give a comprehensive overview and allow a direct comparison of different experiments.
Are the spermatozoa affected by acid and alkaline pH (swimming velocity, acrosome reaction, type of swimming) and can this have an effect on fertilization processes?
Introduction: Lines 111-118 are a summary of results and should be deleted from introduction
Discussion: lines 634 – 658 might better be suited for introduction, lines 658-664 for Material and methods
The study is interesting, the results are of high quality and the different Figures very impressive.
In my opinion the manuscript needs a better organization.
The goal of the study should be clearly justified. Why are oocytes/eggs incubated in acid and alkaline sea water? Has it environmental relevance? On which base pH was selected?
The result section is long, it contains also literature citations (eg. lines 230-232). The legend of Figures is repetitive to the text and could be per se be sufficient as description of the results.
Some data are quantitative and might be reported in this way (elongation of microvilli)
Please report unit in Figures 8a-c. Is this the time scale in msec?
Figure 9 e: What means the combination of pH 8.1 + 9.0?
The authors could think to shorten the text of the results and summarize the major structural and physiological results in form of Table. This kind of presentation could give a comprehensive overview and allow a direct comparison of different experiments.
Are the spermatozoa affected by acid and alkaline pH (swimming velocity, acrosome reaction, type of swimming) and can this have an effect on fertilization processes?
Introduction: Lines 111-118 are a summary of results and should be deleted from introduction
Discussion: lines 634 – 658 might better be suited for introduction, lines 658-664 for Material and methods
Author Response
The study is interesting, the results are of high quality and the different Figures very impressive. In my opinion the manuscript needs a better organization. The goal of the study should be clearly justified. Why are oocytes/eggs incubated in acid and alkaline sea water? Has it environmental relevance? On which base pH was selected?
We thank the reviewer for highlighting the high quality of the Figures presenting our results. As to the question on the goal of the present study, it was clarified in the Introduction of the original version of the manuscript. The rationale for using starfish oocytes in the place of sea urchins was that the oocyte's maturation status (meiosis) is controllable in starfish, unlike in sea urchins. Thus, we decided to study the effect of altered seawater pH on the F-actin-dependent structural reorganization in immature oocytes, maturing oocytes, and fertilized eggs. The pH value of the seawater (6.8 or 9) was the same condition being used previously to study the alteration of the seawater pH (9) on the electrical properties of the plasma membrane and the F-actin-linked morphological changes in sea urchin eggs [ref. 39, 40, 44], leading to an abnormal fertilization response (see Lines 90-106 of the original version of the manuscript).
The result section is long, it contains also literature citations (eg. lines 230-232). The legend of Figures is repetitive to the text and could be per se be sufficient as description of the results.
According to the reviewer's suggestion, we have deleted lines 230-232 containing citations of our previous results using scanning electron microscopy that confirmed the light microscopy observations of the current manuscript. As the reviewer suggested, we rephrased the figure legends more concisely to shorten the length and avoid redundancy.
Some data are quantitative and might be reported in this way (elongation of microvilli)
Unfortunately, a quantitative analysis of microvillar morphology changes (elongation) cannot be performed on the images taken with a transmission electron microscope. This is a limitation of the methodology arising from the obvious fact that the longitudinal section of thin individual microvilli portraying the entire length is nearly impossible for formal quantitative analysis. Quantifying microvillar changes has been done sometimes and published elsewhere by using scanning electron microscopy in sea urchin eggs [ref. 35] whose jelly coats tend to be peeled off spontaneously during fixation. The same thing is technically difficult with starfish.
Please report unit in Figures 8a-c. Is this the time scale in msec?
We have indicated the time scale in the legend of Fig. 8 (A-C) as follows: “The time in the pseudocolor fluorescent images (A-C) indicates minutes and seconds from the standard time format (mm:ss)”.
Figure 9 e: What means the combination of pH 8.1 + 9.0?
We thank the reviewer for asking for clarification about the experimental design of maturing starfish oocytes in seawater at pH 8.1 for 50 minutes and 20 minutes at pH 9. The rationale for performing such experiments was because, at variance with sea urchin eggs fertilized at the end of the maturation process, starfish fertilization occurs when oocytes are undergoing maturation. So, we usually make matured eggs first and then expose and fertilize them in an alkaline seawater condition. In our published studies [ref. 40], sea urchin eggs were exposed for 20 minutes to seawater at pH 9 and then inseminated to investigate the effect of the alkaline seawater on the fertilization response. Since in starfish, the nuclear envelope breakdown didn't take place while the oocytes challenged with the maturing hormone were kept in seawater at pH 9, we decided to stimulate the oocytes in normal seawater (pH 8.1) for 50 minutes to be sure that the intermixing of the nucleoplasm with the cytoplasm normally occurred (GVBD). Then we treated the eggs for extra 20 minutes in alkaline seawater because twenty minutes of exposure to alkaline seawater was already employed for the sea urchin eggs and because 70 minutes of hormonal stimulation is required to achieve optimal fertilizability conditions in this species of starfish. We have added a few lines in the Results to clarify the rationale for performing this experiment.
The authors could think to shorten the text of the results and summarize the major structural and physiological results in form of Table. This kind of presentation could give a comprehensive overview and allow a direct comparison of different experiments.
According to the reviewer’s suggestion, we have summarized the morphological and physiological results providing Table 2.
Are the spermatozoa affected by acid and alkaline pH (swimming velocity, acrosome reaction, type of swimming) and can this have an effect on fertilization processes?
We thank the reviewer for highlighting this point. The results of this contribution have shown that only the induction of the acrosome reaction in sperm exposed to acidic seawater may have been affected, but not their swimming velocity. Sperm arrive at the vicinity of the egg surface with no added time lag. We have inserted this information in the revised manuscript version of the Discussion session.
Introduction: Lines 111-118 are a summary of results and should be deleted from introduction
We have deleted lines 111-118 from the Introduction as suggested by the reviewer.
Discussion: lines 634 – 658 might better be suited for introduction, lines 658-664 for Material and methods
We have moved lines 634-658 to the Introduction and lines 658-664 to the Material and Methods section, as suggested by the reviewer.
Reviewer 2 Report
In this study, the authors examined effects of pH of seawater on fertilization using starfish oocytes. They found that the altered seawater pH affected the number of incorporated sperm in oocytes, fertilization envelope elevation, and Ca2+ increase after fertilization. They also checked changes of the cortical F-actin. This manuscript provides novel and interesting findings. The text is very well written.
Line 202, “For each experimental condition, 5/6 oocytes were used,” should be “For each experimental condition, 5 or 6 oocytes were used”
Line 234 and 267, “insemination cones” may be “fertilization cones”.
In Fig. 2C and Figure S1 B, the number of incorporated sperm into oocytes treated with acidic seawater was fewer than those of control. This effect may be due to a decrease of acrosome reaction inducing substance in jelly, because acrosome reaction inducing substance is extracted in acidic seawater, although significant acrosome reaction inducing substance was still remained in jelly layer as shown in Figure S1 C. Another possibility is that the rate of acrosome reaction depends on pH of seawater, suggesting that low pH inhibits acrosome reaction and fertilization.
Line282, (see Fig. 3 D for comparison) should be (see Fig. 3 F for comparison).
Line 674, Fig. Fig. 3 B should be Fig. 3 B.
Line 724, Another possibility is that GVBD inhibition was due to the effect of the alkaline seawater on the cdk1 activity inducing nuclear lamin phosphorylation.
Author Response
In this study, the authors examined effects of pH of seawater on fertilization using starfish oocytes. They found that the altered seawater pH affected the number of incorporated sperm in oocytes, fertilization envelope elevation, and Ca2+ increase after fertilization. They also checked changes of the cortical F-actin. This manuscript provides novel and interesting findings. The text is very well written.
We thank the reviewer for finding our results novel, interesting and very well written.
Line 202, “For each experimental condition, 5/6 oocytes were used,” should be “For each experimental condition, 5 or 6 oocytes were used”
We thank the reviewer for his suggestion. We have changed the sentence accordingly.
Line 234 and 267, “insemination cones” may be “fertilization cones”.
We understand the request of the reviewer to replace “insemination” with “fertilization” to indicate the sperm-induced cones upon insemination. However, we deliberately used insemination cones produced by immature oocytes challenged with sperm and fertilization cones when mature eggs were fertilized because there is no actual fertilization in the case of immature oocytes. That was the original intention. However, the term ‘insemination cone’ is unprecedented and may cause unnecessary confusion. We have substituted ‘insemination cone’ with ‘fertilization cone,’ as suggested by the reviewer.
In Fig. 2C and Figure S1 B, the number of incorporated sperm into oocytes treated with acidic seawater was fewer than those of control. This effect may be due to a decrease of acrosome reaction inducing substance in jelly, because acrosome reaction inducing substance is extracted in acidic seawater, although significant acrosome reaction inducing substance was still remained in jelly layer as shown in Figure S1 C. Another possibility is that the rate of acrosome reaction depends on pH of seawater, suggesting that low pH inhibits acrosome reaction and fertilization.
We thank the reviewer for asking us to discuss the possibility that the reduced number of sperm entries upon exposure of immature oocytes in acidic seawater may affect the acrosome reaction-inducing substance or the acrosome reaction rate. We have now rephrased the original sentence to enhance the point.
Line282, (see Fig. 3 D for comparison) should be (see Fig. 3 F for comparison).
We thank the reviewer for picking up our mistake. We have replaced Fig. 3 D with Fig. 3 F.
Line 674, Fig. Fig. 3 B should be Fig. 3 B.
We have deleted the redundant “Fig”. as suggested by the reviewer.
Line 724, Another possibility is that GVBD inhibition was due to the effect of the alkaline seawater on the cdk1 activity inducing nuclear lamin phosphorylation.
We thank the reviewer for suggesting the possibility that alkaline seawater inhibits GVBD by blocking the cdk1 activity. We have added this possibility to the Discussion session.
Reviewer 3 Report
Effect of the Acidic and Alkaline Seawater on the F-actin-dependent Ca2+ Signals following Insemination of Immature Starfish Oocytes and Mature Eggs
By Nunzia Limatola, Jong Tai Chun, Suzanne C. Schneider, Jean-Louis Schmitt, Jean-Marie Lehn and Luigia Santella
The article evaluates the effects of different seawater pH on Ca signals after insemination of immature starfish oocyte. The results are very interesting and show that the maturation process and the dynamic structural changes of the cortical F-actin depends on pH, altering fertilization and sperm penetration. Data shown are clear.
1. It is important to adress the physiological implications about changes in pH if they occur in natural conditions and what ambiental factors could alter seawater pH.
Author Response
The article evaluates the effects of different seawater pH on Ca signals after insemination of immature starfish oocyte. The results are very interesting and show that the maturation process and the dynamic structural changes of the cortical F-actin depends on pH, altering fertilization and sperm penetration. Data shown are clear.
We were pleased that the reviewer found our results clear and interesting.
- It is important to adress the physiological implications about changes in pH if they occur in natural conditions and what ambiental factors could alter seawater pH.
We have added a few lines to discuss the importance of investigating the effect of seawater changes on the physiological response of marine organisms that reproduce externally at sea under controlled experimental conditions in the laboratory. The experimental conditions used in this study are closer to the extreme conditions that would be hardly encountered even in this epoch of climate change, etc. However, the same experimental conditions (pH 6.8 and 9) have been employed previously in studying the eggs from other species that reproduce similarly, namely sea urchins. Hence, the merit of our study using the identical condition might be that it can make the comparisons possible not only with other echinoderm eggs but also with other cell types encountered in the research of cell biology. This point on environmental relevance has been elaborated on in the revised manuscript.
Round 2
Reviewer 1 Report
No further comments. I recommend publication of the manuscript.